# A Hierarchical and Abstraction-Based Blockchain Model

**Swagatika Sahoo[1], Akshay M. Fajge [1], Raju Halder [1] and Agostino Cortesi [2],***

[1]   Department of Computer Science and Engineering, Indian Institute of Technology Patna, Bihta, Patna 801106, Bihar, India; swagatika_1921cs03@iitp.ac.in (S.S.); fajge_1921cs12@iitp.ac.in (A.M.F.); halder@iitp.ac.in (R.H.)

[2]   Dipartimento di Scienze Ambientali, Informatica e Statistica, Università Ca' Foscari, via Torino 155, 30170 Venice, Italy

*   Correspondence: cortesi@unive.it

**Abstract:** In the nine years since its launch, amid intense research, scalability is always a serious concern in blockchain, especially in case of large-scale network generating huge number of transaction-records. In this paper, we propose a hierarchical blockchain model characterized by: (1) each level maintains multiple local blockchain networks, (2) each local blockchain records local transactional activities, and (3) partial views (tunable w.r.t. precision) of different subsets of local blockchain-records are maintained in the blockchains at next level of the hierarchy. To meet this objective, we apply abstractions on a set of transaction-records in a regular time interval by following the Abstract Interpretation framework, which provides a tunable precision in various abstract domain and guarantees the soundness of the system. While this model suitably fits to the real-worlds organizational structures, the proposal is powerful enough to scale when large number of nodes participate in a network resulting into an enormous growth of the network-size and the number of transaction-records. We discuss experimental results on a small-scale network with three sub networks at lower-level and by abstracting the transaction-records in the abstract domain of intervals. The results are encouraging and clearly indicate the effectiveness of this approach to control exponential growth of blockchain size w.r.t. the total number of participants in the network.

**Keywords:** blockchain; abstract interpretation; hierarchical model

## 1. Introduction

Since the pioneer work by Satoshi Nakamoto in 2008 [1], Blockchain has emerged as a revolutionary technology with numerous applications in practice, including cryptocurrency, food safety, IoT security, healthcare, and many more [2,3]. Blockchain is essentially an append-only data structure (in the form of sequence of blocks) containing complete list of transaction-records, like conventional public ledger, which is maintained by multiple non-trusting members connected via a decentralized peer-to-peer network. Such distributed ledger provides a powerful mean to verify records, without any trusted intermediary such as brokers, agents, etc. [4,5].

Although blockchain technology has great potential to transform most industries, but at this nascent stage it is facing a number of technical challenges. The most serious challenge is scalability [6,7]. In the age of big data, as blockchain network size increases day by day, the amount of transaction-records generated in the network also increases rapidly. This yields a growth of blockchain size at the same pace due to its append-only tamper-evident structure. Managing such huge sized blockchain by the network nodes in a distributed setting is of course a matter of concern. This may gradually lead to a centralized situation as less users would like to maintain such a large blockchain [8]. For example,

the size of the Bitcoin blockchain has been growing since its creation in 2009, reaching approximately 197 gigabytes in size by the beginning of January 2019. Currently the Bitcoin network is able to process nearly 7 transactions per second due to its block-size limitation upto 1 MB with a 10 min mining gap [9]. This clearly indicates the inability of dealing high frequency trading, compared to PayPal or visa. On the other hand, increase of block size limit may cause propagation delay in the network and may lead to branches in the blockchain [10]. As more users incur more transactions it takes longer for verification with waiting times increasing sharply at peak times, and thereby reduces the operational efficiency gradually in the network.

Furthermore, as most organizational structures are hierarchical in general, this causes a severe limitation when to adopt blockchain-based solution in order to fulfill organizational needs. Although private blockchain may protect information from outsider, however there is no specific mechanism to impose access control enabling data-access differently to different groups across the hierarchy within the same organization. The only possible way to enforce access control is to deploy smart contract specifying policy specification, which is of course a tedious and error-prone task [11,12].

To ameliorate such bottlenecks, in this paper, we propose a hierarchical blockchain model where each level maintains multiple local blockchain-networks, each of which records local transactional activities, and partial views (tunable w.r.t. precision) of different subsets of local blockchain-records are maintained in the blockchains at next level of the hierarchy. Figure 1 depicts a large-scale network in a hierarchical fashion involving stakeholders across the globe. The green circles indicate lowest level blockchain networks, while the red circles indicate data management at a higher level of the hierarchy.

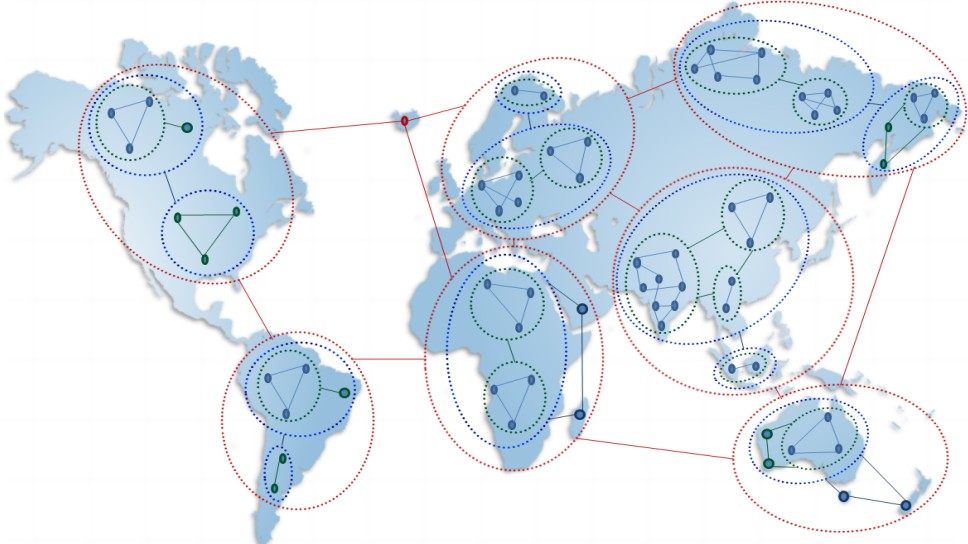

**Figure 1.** Decentralized network views at various levels of hierarchy.

While this makes the system scalable enough by allowing at each level multiple local blockchains (rather than single one at a global level) as a way to reduce overall network activities, alongside such views also facilitate to control the access of blockchains (and hence blockchain records) differently to different groups of people or processes as per their access rights. To this aim, we extend the Abstract Interpretation framework [13] to provide a sound approximation of blockchain-records across the hierarchical levels, making them tunable w.r.t. precision in various abstract domains ranging from non-relational to relational. The use of Abstract Interpretation framework in our approach guarantees soundness of blockchain operations (for example, Auditability) due to systematic constructions of abstractions following Galois connection under the framework. In particular, we address the following two important aspects: (1) architecture of the networking model by moving from a mesh to a hierarchical structure, and (2) precision control of blockchain-records across the levels of the hierarchy by abstracting sets of data.

The prime intuition behind our proposal is to deal locally-generated information separately in the form of local blockchains which reduces performance bottleneck significantly, and to abstract these local blockchain-records in the next level of the hierarchy with precision control. For instance, if a travel agency H reserves seats 1A, 1B, 1C, 2A, 2B, 2C, 3A, 3C for personnel of the same firm N, instead of adding to the ledger all eight transactions <H, N, 1A>, ..., <H, N, 3C>, an abstract representation like <H, N, [1,2][A,B], [1,3][A,C]> may under- and over-approximate this set of transactions. Although blockchain provides a transparent mean to validate transactions, there are several scenarios where a number of participants in the network do not require exact information and the answer in the form of "yes" or "no" is sufficient during the verification process. For instance, in case of ledger containing insurance records, traffic police may expect a decisive answer when he/she verifies if a person is insured with an amount in a specific range.

A schematic diagram of our proposed model organizing blockchains in a hierarchical fashion is depicted in Figure 2, where $BC_i$ represents blockchains at the hierarchy level $i$. Observe that $BC_2$ contains an abstract representation (partial view) of all the transaction-records in the blockchains $BC_1$.

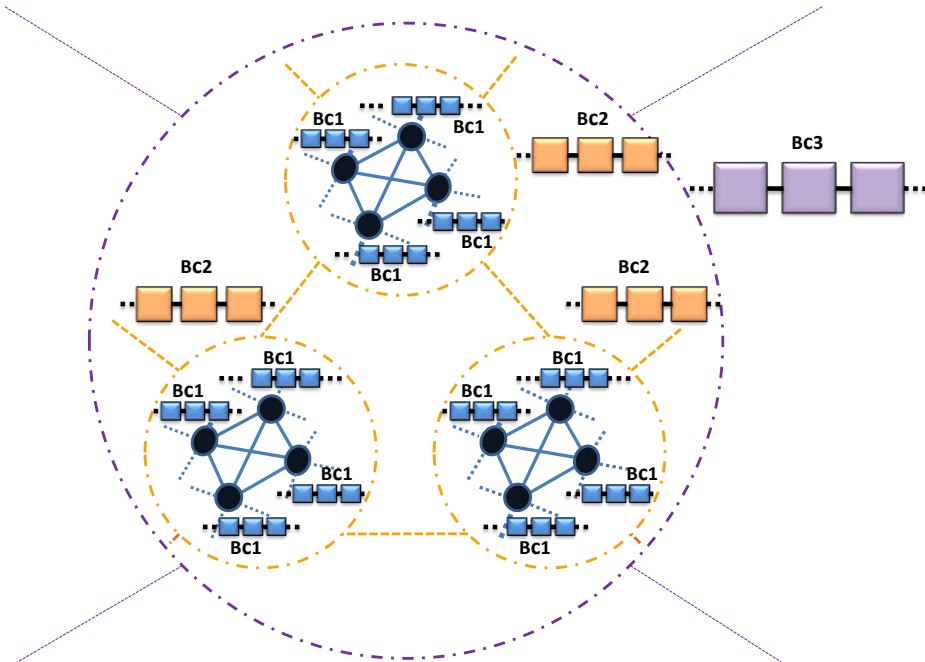

**Figure 2.** Schematic diagram of hierarchical blockchains model.

To summarize, the main contributions in this paper are:

- We propose a hierarchical blockchain model as a way to improve scalability in case of large-scale blockchain network.
- The proposal involves data abstractions across the levels of the hierarchy, by extending the Abstract Interpretation theory [13]. This ensures the soundness of the blockchain operations, with a precision control at different levels of abstractions.
- We illustrate how such model fits into real-world organizational scenarios, providing access control differently to different groups of people or processes in a system as per their access rights.
- Finally, we implement the proposal by considering only interval abstraction, and we demonstrate the results showing the variation of blockchain size at different hierarchical level w.r.t. mining delay.

The structure of the paper is organized as follows: Section 2 illustrates a motivating example. Section 3 discusses related works in the literature. We recall the basics on the Abstract Interpretation theory in Section 4. The detail proposal on the hierarchical blockchain model is described in Section 5. An implementation of our proposed approach and the experimental results in the domain of intervals

are reported in Section 6. We provide a detail discussions on the proposal w.r.t. the literature in Section 7. Finally, we conclude our work in Section 8.

## 2. A Motivating Example

Let us consider a multinational insurance company which is expanded over a number of countries across the globe. The company is associated, directly or indirectly, with many stakeholders, including its own employees, country's road and transport department, traffic polices, empanelled hospitals, common people, and many others. To facilitate operations like insurance record verification, insurance claims, etc., researchers have already proposed a number of blockchain-based solutions [14–16]. This is to observe that, in all these cases, the proposed systems are somehow dependent on either local database or third-party cloud storage to store insurance data. This is due to the fact that maintenance of a single blockchain and populating it with large volume of insurance data which is generated everyday across the globe may eventually slowdown the performance of the whole system, as its size grows.

Interestingly, there are many situations when exact details of the data may not be required to answer queries issued by the stakeholders. For example, a traffic police needs to verify whether a car owner has valid insurance, an insurance company needs to verify whether the person already has Mediclaim policy with any other company, etc. A common objective in all such cases is to obtain a decisive answer in the form of "yes" or "no" and the presence of insurance data in an abstract form (representing specific properties of interest) in the blockchain fulfills such functional requirements. This leads to a significant reduction of the overall size of the blockchain in the system, improving its performance thereof.

Inspired from these facts, in the subsequent sections, we propose a hierarchical blockchain model where multiple blockchains at different levels representing data in various forms, from concrete (lowest level of the hierarchy) to abstract (towards higher levels), are considered. Such model may act as a powerful solution to deal with scalability, access control, data sharing, etc.

## 3. Related Works

Blockchain over recent years has been extolled as a revolution in business technology [17]. In the nine years since its launch, an intensive research in the field has proved its immense potential in a wide range of applications, ranging from new land registries, KYC applications, e-commerce, insurance, supply chain, and many more [18]. The most impressive results have seen blockchains used to store information, cut out intermediaries, and enable greater coordination between companies, for example in relation to data standards. Detail survey on blockchain covering the technology along with its possible applications and challenges are found in [6,19,20].

In this section, we mainly restrict our discussion to the scalability issue of blockchain. The authors in [6] identified various challenges which create major hindrance in the future pathway of blockchain technology. They emphasize the scalability issue as one such concern when to deal with large-scale decentralized network and big data. As a solution, they refer two different approaches: Storage optimization of blockchain [21] and Redesigning blockchain [22]. In the first approach, the author suggested to prune out old transactions from the blockchain network to deal with scalability. Only information about non-empty transactions are maintained, offering benefits such as faster transactions, lower fees, more blockspace, faster network synchronization, etc. The second approach [22] introduces a new blockchain protocol, Bitcoin-NG (Next Generation), aiming to scale bitcoin network, with bandwidth limited only by the capacity of the individual nodes and latency limited only by the propagation time of the network.

The authors in [23] proposed a new blockchain storage "Mystiko" which is built over Apache Cassandra database. The main aim is to deal with big data. The authors claims that Mystiko supports the important properties like transaction throughput, high scalability, high availability and other text retrieval features. We observed that, although the support of distributed storages (like Cassendra) successfully stores bigdata in blockchain, however they need highly equipped nodes in the network.

Two other works which also support such kind of architecture are BigchainDB which is built on MongoDB [24] and HBasechainDB which is based on HBase with Hadoop [25].

The authors in [26] introduced the notion of overlay network where a number of chosen cluster head maintains a separate blockchain, in addition to the local blockchains. The main objective of this proposal is to ensure IoT privacy and security by eliminating the overhead of computationally expensive blockchain management due to high bandwidth requirement and possible delays. Although the approach is quite similar to our case, but the network structure is only limited to two-level hierarchy and therefore it is not scalable enough in practice. Paper [27] addressed the issue of data boundary conflicts in Land Registration using blockchain. The Land Registry and Cadastre (LR & C) blockchain database component of a node considers three levels in a hierarchy. This is immediate to claim that the proposed architecture is very specific to this application and does not address scalability. Recently another related work is proposed in [28] which focuses exclusively IoT systems that keep user's key information, and hence may not fit to other scenario in general. They discriminate the participants, categorizing into device-layer, fog-layer and cloud-layer nodes, with a varying computational capabilities. The task of each layer nodes are predefined. For Example, the fog-layer nodes which lie in between device-layer and cloud layer, act as security manager to record and verify transactions that include key management information in this respective domain. Cloud-nodes are highly capable to manage and function multiple blockchains. Although the proposal is suitable for cross domain communications respecting privacy issues, it does not address any scalability issue in presence of large-scale nodes in the peer-to-peer network.

In [7], the author introduced a distributed access control model for blockchain in case of large-scale IoT systems consisting of billions devices over the globe. The access control policy is defined in smart contract deployed in the blockchain. The important components facilitating this access control in a distributed way are the introduction of access control manager nodes (light weight) and management hub nodes (high computation nodes) in the network. Observe that the model is suitable only for private blockchain network. As nodes can easily participate in the network globally, it may be a cumbersome and error-prone task to define access control policy for the large numbers of IoT devices.

There are several other proposals which address access control in blockchain network by incorporating access-control policies in the form of smart contracts [11,12,29]. However, in these proposals, all nodes form a single level blockchain network.

## 4. Preliminaries: Abstract Interpretation

Abstract Interpretation [13] is a mathematical framework, originally introduced by Cousot and Cousot in 1977, which provides a sound approximation of programs concrete semantics enabling sound answers to questions about programs run-time behaviors. The framework has been applied to different applications areas, from Security (e.g., [30,31]) to Databases (e.g., [32,33]), and to different programming environments (e.g., [34–36]). The idea is to lift concrete semantics to an abstract setting by replacing concrete values by suitable properties of interest, and simulating the concrete operations by sound abstract operations. The concrete and the abstract domains are always partial orders, where the ordering relation describes the relative precision of the denotations and where the top element represents no information. The mapping between concrete and abstract semantics domains is established by Galois connection:

**Definition 1** (Galois connections [13])**.** *Consider two partial orders* $(\mathbb{D}, \leqslant)$ *and* $(\overline{\mathbb{D}}, \sqsubseteq)$ *where the first one represents a concrete domain and the second one represents an abstract domain. The Galois connection between* $\mathbb{D}$ *and* $\overline{\mathbb{D}}$ *is denoted by* $\left\langle (\mathbb{D}, \leqslant), \alpha, \gamma, (\overline{\mathbb{D}}, \sqsubseteq) \right\rangle$ *or* $(\mathbb{D}, \leqslant) \xleftarrow{\alpha}_{\gamma} (\overline{\mathbb{D}}, \sqsubseteq)$ *where* $\alpha \colon \mathbb{D} \to \overline{\mathbb{D}}$ *and* $\gamma \colon \overline{\mathbb{D}} \to \mathbb{D}$ *holds iff:*

- $\forall v \in \mathbb{D}. \ v \sqsubseteq \gamma \circ \alpha(v).$
- $\forall \overline{v} \in \overline{\mathbb{D}}. \ \alpha \circ \gamma(\overline{v}) \sqsubseteq \overline{v}.$
- $\alpha$ *and* $\gamma$ *are monotonic.*

*In other words, iff* $\forall v \in \mathbb{D}, \overline{v} \in \overline{\mathbb{D}}. \; \alpha(v) \sqsubseteq \overline{v} \iff v \leqslant \gamma(\overline{v}).$

For example, given a domain of integers $Z$, we can establish a Galois connection $\big\langle (\wp(Z), \subseteq),$ $\alpha, \gamma, (\text{SIGN}, \sqsubseteq) \big\rangle$ between the concrete domain $\wp(Z)$, the powerset of $Z$, and an abstract domain SIGN=$\{\top, +, -, 0, \pm, 0+, 0-, \bot\}$ representing sign properties of numerical values, depicted in Figure 3.

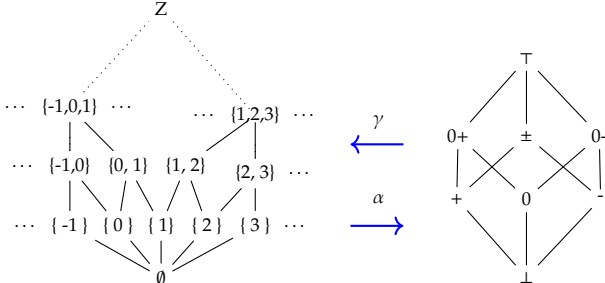

**Figure 3.** Galois connection between $\wp(Z)$ and SIGN.

The abstraction function $\alpha$ and concretization function $\gamma$ between the domains are defined below:

$$\forall X \in \wp(Z): \; \alpha(S) = \begin{cases} \bot & \text{if } S = \emptyset \\ + & \text{if } S = \{\, a \mid a > 0 \,\} \\ 0 & \text{if } S = \{\, 0 \,\} \\ - & \text{if } S = \{\, a \mid a < 0 \,\} \\ \pm & \text{if } S = \{\, a \mid a > 0 \; \vee a < 0 \,\} \\ 0+ & \text{if } S = \{\, a \mid a \geqslant 0 \,\} \\ 0- & \text{if } S = \{\, a \mid a \leqslant 0 \,\} \\ \top & \text{Otherwise} \end{cases}$$

$$\forall \overline{v} \in \text{SIGN}: \gamma(\overline{v}) = \begin{cases} \emptyset & \text{if } \overline{v} = \bot \\ \{k \in Z \mid \; k > 0\} & \text{if } \overline{v} = + \\ \{0\} & \text{if } \overline{v} = 0 \\ \{k \in Z \mid \; k < 0\} & \text{if } \overline{v} = - \\ \{k \in Z \mid \; k > 0 \vee k < 0\} & \text{if } \overline{v} = \pm \\ \{k \in Z \mid \; k \geqslant 0\} & \text{if } \overline{v} = 0+ \\ \{k \in Z \mid \; k \leqslant 0\} & \text{if } \overline{v} = 0- \\ Z & \text{Otherwise} \end{cases}$$

The arithmetic operations over the abstract domain are defined accordingly, ensuring the soundness w.r.t. their concrete counter-part [37]. For example, the '$\times$' operation over the concrete domain is mapped to its abstract version '$\times^{\sharp}$' as follows: $-(\times^{\sharp})- = +, \; +(\times^{\sharp})- = -, \; +(\times^{\sharp})+ = +, \; \top(\times^{\sharp})+ = \top,$ $\bot(\times^{\sharp})+ = \bot$, and so on.

A number of abstract domains, non-relational and relational, exist in the literature [13,37–40].

### 4.1. Non-Relational Abstract Domains

An abstract domain is said to be non-relational if it does not preserve any relations among program variables. Non-relational abstract domains care only about the actual variables being updated, rather than having potential to change multiple values at once [37]. Some widely used non-relational abstract domains for program analysis include sign domain for sign property analysis, parity domain for parity property analysis, interval domain for division-by-zero or overflows, etc. [13]. Analysis in these domains, although effective, is less precise w.r.t. relational abstract domains. Figure 4 pictorially

depicts a scenario where a set of points SP (indicated by ●) on the xy-plane are abstracted by sign and interval properties.

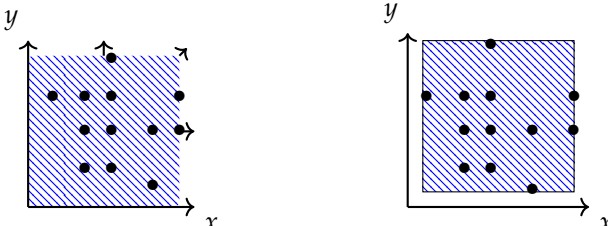

**Figure 4.** Abstraction of SP by Sign and Interval Properties.

### 4.2. Relational Abstract Domains

Unlike non-relational abstract domains, the relational abstract domains preserve relations among program variables [38]. Analysis in these domains are more precise as compared to the non-relational abstract domains, in particular when more number of relations among variables are present in the code itself. Widely used relational abstract domains are the domains of Polyhedra, Octagons, Difference-Bound Matrices (DBM), etc. [38–40]. Abstraction of the same set of points in Octagon and Polyhedra domains are exemplified in Figure 5.

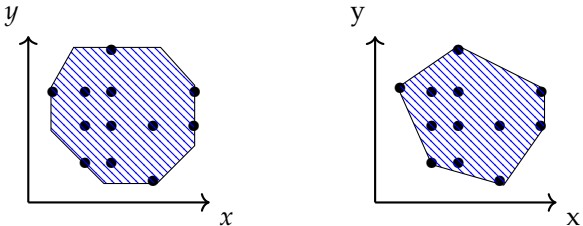

**Figure 5.** Abstraction of SP in Octagon and Polyhedra Domains.

## 5. Hierarchical Blockchain Model

In this section, we detail our proposal. To do so, we first describe an abstraction of concrete blocks in a blockchain and then we describe the mining process of the abstract blocks to the blockchain in the next hierarchical level.

### 5.1. Block Structure and Data Abstraction

In general, information in blocks of a blockchain are categorized into the followings:

- Block-specific Information: This information includes block identifier, version, hash of the block header, etc.
- Block-header Information: This section of information contains hash pointer of the previous block, mining details (such as difficulty level, nonce, etc.), time-stamp, Merkle tree root, etc.
- Transaction-specific Information: This information includes all transaction-data, along with its associated Merkle tree details (except its root).

Transaction-data may have different semantics depending on its application domain. One may use different data structure (e.g., bucket [41]) for transaction-records for the optimization of blockchain-structure. An example of a generic block structure is depicted in Figure 6.

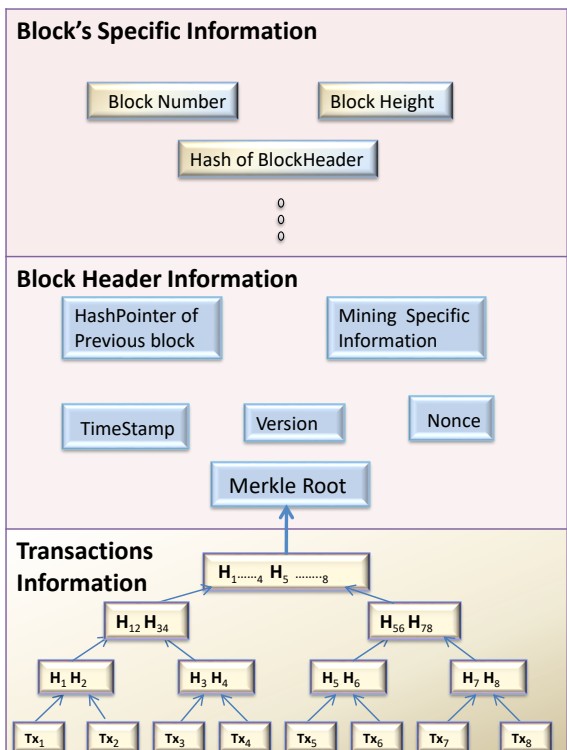

**Figure 6.** A Generic Block Structure.

Let us now describe how transaction-data (numerical, string, etc.) in blocks can be abstracted, enabling one to design blockchains at various levels of abstractions with precision tuning.

### 5.1.1. Blockchain in Interval Domain

The interval abstract domain [13] is defined by $\mathbb{I} = \big\{[l, h] \mid l \in \mathbb{Z} \cup \{-\infty\}, h \in \mathbb{Z} \cup \{+\infty\}, l \leq h\big\} \cup \bot$. Given a set of integers X, this domain approximates X by a pair $[l, h]$ where $l$ and $h$ represents minimal and maximal element in X respectively. For example, the set $\{2, 1, 100, 4\}$ is approximated by $[1, 100]$.

Let $\mathsf{L}_c = \langle \wp(\mathbb{R}), \subseteq, \emptyset, \mathbb{R}, \cap, \cup \rangle$ and $\mathsf{L}_\mathbb{I} = \langle \mathbb{I}, \sqsubseteq, \bot, [-\infty, +\infty], \sqcap, \sqcup \rangle$ be the concrete lattice of the powerset of numerical values $\mathbb{R}$ and the abstract lattice corresponding to the domain of intervals $\mathbb{I}$ respectively, where

$$[l_1, h_1] \sqsubseteq [l_2, h_2] \iff l_2 \leqslant l_1 \wedge h_2 \geqslant h_1$$
$$[l_1, h_1] \sqcap [l_2, h_2] = [max(l_1\ l_2),\ min(h_1\ h_2)]$$
$$[l_1, h_1] \sqcup [l_2, h_2] = [min(l_1,\ l_2),\ max(h_1\ h_2)]$$

The correspondence between $\mathsf{L}_c$ and $\mathsf{L}_\mathbb{I}$ is defined by Galois connections $\langle \mathsf{L}_c, \alpha_\mathbb{I}, \gamma_\mathbb{I}, \mathsf{L}_\mathbb{I} \rangle$, depicted in Figure 7, where $\forall S \in \wp(\mathbb{Z})$ and $\forall \overline{v} \in \overline{\mathbb{I}}$:

$$\alpha_\mathbb{I}(S) = \begin{cases} \bot & \text{if } S = \emptyset \\ [l, h] & \text{if } round\_down(min(S)) = l \wedge round\_up(max(S)) = h \\ [-\infty, h] & \text{if } \nexists min(S) \wedge round\_up(max(S)) = h \\ [l, +\infty] & \text{if } round\_down(min(S)) = l \wedge \nexists max(S) \\ [-\infty, +\infty] & \text{if } \nexists min(S) \wedge \nexists max(S); \end{cases}$$

$$\gamma_{\mathbb{I}}(\overline{v}) = \begin{cases} \emptyset & \text{if } \overline{v} = \bot \\ \{k \in \mathbb{R} \mid l \le k \le h\} & \text{if } \overline{v} = [l, \, h] \\ \{k \in \mathbb{R} \mid k \le h\} & \text{if } \overline{v} = [-\infty, \, h] \\ \{k \in \mathbb{R} \mid l \le k\} & \text{if } \overline{v} = [l, \, +\infty] \\ \mathbb{R} & \text{if } \overline{v} = [+\infty, \, -\infty]. \end{cases}$$

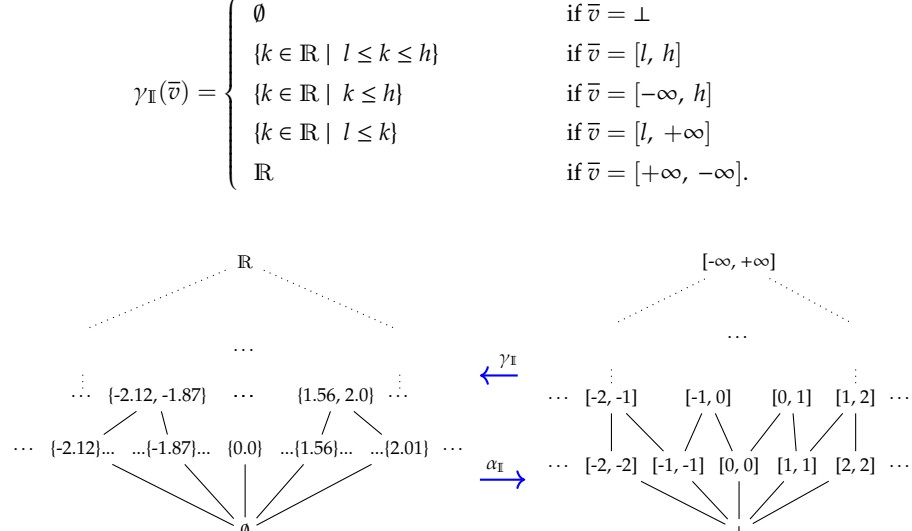

**Figure 7.** Galois connection between $\mathsf{L}_c$ and $\mathsf{L}_{\mathbb{I}}$.

Given a concrete block, its corresponding abstract version by replacing all concrete information with intervals is depicted in Figure 8:

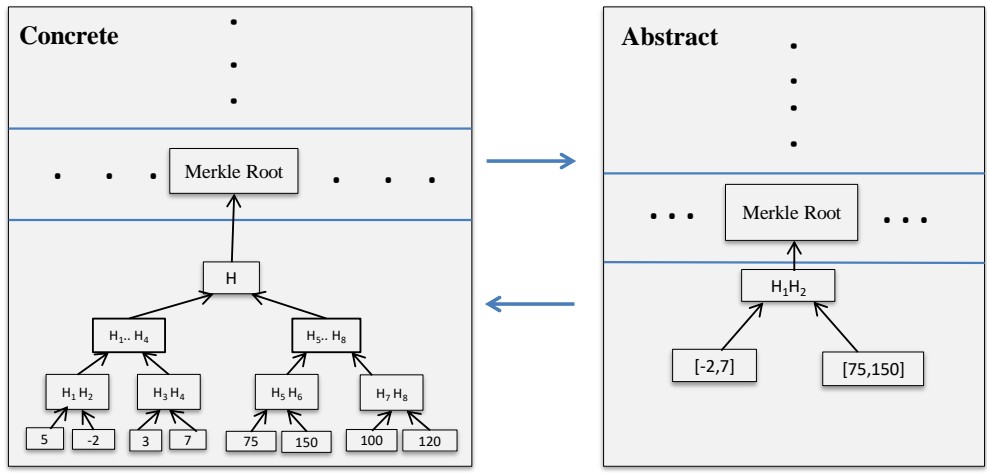

**Figure 8.** Abstract Block in Interval Domain.

## 5.1.2. Blockchain in String Abstract Domain

In this section, we present abstract domains for string data. Although various abstract domains exist, let us recall from [42] two among them: "Prefix and Suffix" and "Bricks". Given an alphabet $\Sigma$ consisting of characters, a string is a sequence (possibly infinite) of elements in $\Sigma$. Let us consider $S = \Sigma^*$, which is the set of all possible strings of any length (possibly infinite) generated from $\Sigma$. Formally, the concrete string domain is represented in the form of the lattice: $\mathsf{L}_c = \langle \wp(S), \subseteq, \emptyset, S, \cup, \cap \rangle$.

### Prefix and Suffix Abstract Domain

A prefix $x$ is an abstraction of a set of strings (including the string $x$ itself) which begin with $x$ followed by other characters. Formally, the prefix abstract domain is represented by the abstract lattice $\mathsf{L}_{\mathbb{A}} = \langle \mathbb{A}, \sqsubseteq, \bot, \top, \sqcup, \sqcap \rangle$ where $\mathbb{A} = \Sigma^* \cup \{\bot\}$. Given two prefixes $\overline{x}, \overline{y} \in \mathbb{A}$, the partial order $\sqsubseteq$ is defined as:

$$\overline{x} \sqsubseteq \overline{y} \quad \text{iff} \quad len(\overline{x}) \ge len(\overline{y}) \land \forall i \in [0, len(\overline{y}) - 1] : \overline{x}[i] = \overline{y}[i]$$

where $len(a)$ returns the number of characters in '$a$'. In other words, $\overline{y}$ is greater than $\overline{x}$ iff $\overline{y}$ is the prefix of $\overline{x}$. Observe that the top element $\top$ in $\mathbb{A}$ is the empty prefix $\epsilon$. The meet operation $\sqcap$ and join operation $\sqcup$ in is defined as:

$$\sqcap(\overline{x_1}, \overline{x_2}) = \begin{cases} \overline{x_1} & \text{if } \overline{x_1} \sqsubseteq \overline{x_2} \\ \overline{x_2} & \text{if } \overline{x_2} \sqsubseteq \overline{x_1} \\ \bot & \text{Otherwise} \end{cases}$$

$$\sqcup(\overline{x_1}, \overline{x_2}) = \begin{cases} \overline{x_1} & \text{if } \overline{x_2} \sqsubseteq \overline{x_1} \\ \overline{x_2} & \text{if } \overline{x_1} \sqsubseteq \overline{x_2} \\ \top & \text{Otherwise} \end{cases}$$

Therefore, the Galois connection between $\mathsf{L}_c$ and $\mathsf{L}_\mathbb{A}$ is established as $\langle \mathsf{L}_c, \alpha_\mathbb{A}, \gamma_\mathbb{A}, \mathsf{L}_\mathbb{A} \rangle$, where

$$\forall X \in \wp(S): \quad \alpha_\mathbb{A}(X) = \sqcup_{x \in X} \alpha(x) \quad \text{and} \quad \alpha(x) = x$$

and

$$\forall \overline{v} \in \mathbb{A}: \quad \gamma_\mathbb{A}(\overline{v}) = \{s \in S \mid \text{prefix}(s) = \overline{v}\}$$

The suffix domain is constructed in the similar way. Given a concrete block, its corresponding abstract version in the prefix domain is depicted in Figure 9.

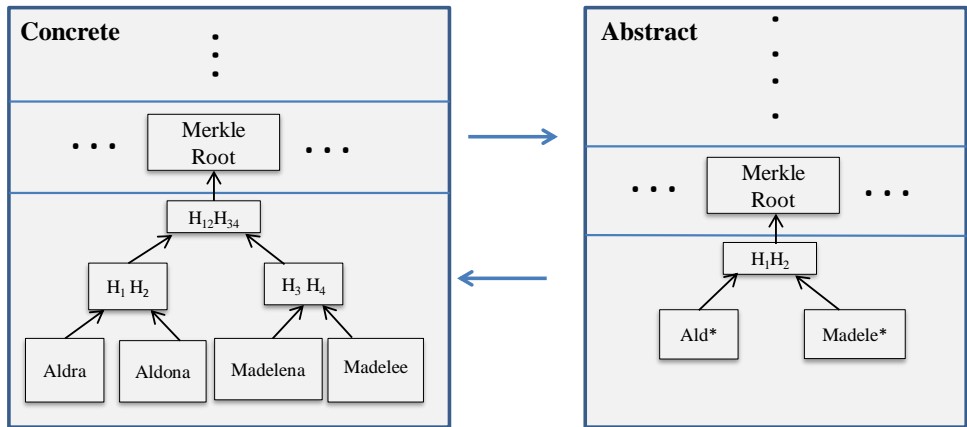

**Figure 9.** Abstract Block in Prefix Domain.

Bricks Abstract Domain

Bricks abstract domain [42] is defined with an aim to capture both inclusion and order amongst characters, using regular expressions [43]. A brick is defined as an element of the form $\overline{B} = [X]^{min,max}$ where $X \in \wp(S)$. Both *min* and *max* are positive integer values which represent that all strings are built by combining (in order) the strings in $X$ taking each of them between *min* and *max* times altogether. For example, $[\{\text{"so"}, \text{"le"}\}]^{1,2} = \{so, le, soso, lele, sole, leso\}$.

In bricks abstract domain, a set of strings is approximated by a combination of bricks. For example, $[\{\text{"manage"}\}]^{0,1} [\{\text{"ment"}\}]^{1,1} = \{ment, management\}$, where $[\{\text{"manage"}\}]^{0,1}$ concretizes to $\{\epsilon, \text{"manage"}\}$ and $[\{\text{"ment"}\}]^{1,1}$ concretizes to $\{\text{"ment"}\}$.

Formally, the abstract domain of bricks is defined as: $\overline{BR} = \overline{B}^*$; that is, the set of all finite sequences of bricks. The abstract partial order in bricks domain is $\langle \overline{BR}, \sqsubseteq, \bot, \top, \sqcap, \sqcup \rangle$. The partial order $\sqsubseteq_{\overline{B}}$ on two bricks $[X_1]^{min_1,max_1}, [X_2]^{min_2,max_2} \in \overline{B}$ is defined as $[X_1]^{min_1,max_1} \sqsubseteq_{\overline{B}} [X_2]^{min_2,max_2} \iff (X_1 \subseteq X_2 \wedge min_1 \geq min_2 \wedge max_1 \leq max_2) \vee [X_2]^{min_2,max_2} = \top_{\overline{B}} \vee [X_1]^{min_1,max_1} = \bot_{\overline{B}}$. The partial order $\sqsubseteq$ between two lists of bricks $\overline{L}_1, \overline{L}_2 \in \overline{BR}$ is defined as follows: $\overline{L}_1 \sqsubseteq \overline{L}_2 \iff (\overline{L}_1 = \bot \vee \overline{L}_2 = \top \vee \forall i \in [1 \ldots n] : \overline{L}_1[i] \leq \overline{L}_2[i])$. An upper bound operator is defined as $\sqcup(\overline{L}_1, \overline{L}_2) = \overline{L}_R[1]\overline{L}_R[2] \ldots \overline{L}_R[n]$

where $\forall i \in [1 \ldots n]$: $\overline{L}_R[i] = \sqcup_{\overline{B}}(\overline{L}_1[i], \overline{L}_2[i])$ and $\sqcup_{\overline{B}}$ is defined as $\sqcup_{\overline{B}}([X_1]^{min_1,max_1}, [X_2]^{min_2,max_2}) = [X_1 \cup X_2]^{minimum(min_1,min_2),maximum(max_1,max_2)}$. Similarly, a lower bound operator $\sqcap$ is defined. This way, blockchains can be designed in an abstract domain at various precisions levels. Although we have shown abstract blockchains in non-relational abstract domains only, relational abstractions, such as octagons, polyhedra, etc., may be used in specific contexts.

**Observation**: Processing speed in abstract blockchain may vary depending on the degree of abstraction used. For example, the use of relational abstract domains such as polyhedra domain, although more precise, may exhibits high computation cost ($O(2^n)$ for $n$-dimensional data). On the other hand, the use of non-relation domains such as an interval domain may result into high processing speed ($O(n)$) by compromising precision of the transaction-records. The proposed model supports the use of different abstract domains in different blockchain networks across the hierarchy.

*5.2. Horizontal vs. Vertical Abstraction*

In case of thin clients or lightweight nodes with limited resources, the nodes only store a piece of data only relevant to them, rather than storing the entire blockchain [9]. For example, the piece of information may include only block headers and transactions relevant to the nodes. In other words, such slice of information can be viewed as a vertical abstraction. The limitation is that these kind of nodes do not contribute to the security of the system and they heavily depend on other fully validating nodes.

In contrast to the vertical abstraction, our proposal can be considered as a kind of horizontal abstraction where multiple records are abstracted resulting into a reduction of the number of blocks in the blockchain.

A hybrid abstraction can also be constructed combining the above two when to deal with scalability covering large geographical area with a number of thin clients involved in the system.

*5.3. Consensus Mechanism*

Let $\text{BC}_l^h$ be a blockchain maintained by $l$-th peer-to-peer network $\text{PP}_l^h$ at hierarchy level $h$, where $l = 1, \ldots, n_h$. At lower level $h = 1$, the set of networks is $\text{PP}^1 = \{\text{PP}_1^1, \text{PP}_2^1, \ldots, \text{PP}_{n_1}^1\}$. Given $k$-th subset $S_k \subseteq \text{PP}^1$, one representative member (who acts as leader) from each network in $S_k$ together forms a network $\text{PP}_k^2$ at the next level of hierarchy $h = 2$, where $k = 1, \ldots, n_2$. Therefore, at level $h = 2$, the set of networks is $\text{PP}^2 = \{\text{PP}_1^2, \text{PP}_2^2, \ldots, \text{PP}_{n_2}^2\}$. Following similar approach, networks at subsequent levels are formed. Each network follows traditional consensus algorithm in building and maintaining local blockchain.

Role of Leader Participating in Multilevel Blockchain

Multilevel blockchains are useful in dealing with large volume of data generated across the globe. Since leaders from a subset of networks at level $h$ is also participating into another network at level $h + 1$, such nodes are fully responsible to maintain data-consistency among blockchains across the levels. To preserve data semantics, a leader should listen a number of transactions $\{\tau_i\}$ (which are mined in the blockchain at level $h$) for wait-time $\delta$ and then broadcast a transaction $\overline{\tau}$ (which is a sound approximation of $\{\tau_i\}$) in the $(h + 1)$-level network. However, $\delta$ can be varied to obtain different precision depending on the applications or the network latency and throughput.

The most important feature in such multilevel hierarchy is the ability to ensure soundness of the information as we move towards the higher level of the hierarchy compromising the precision of information. Consider $\text{BC}_l^{h_i}$ and $\text{BC}_k^{h_j}$ such that $h_i \leq h_j$ and there exists a common node (leader) participating in both the networks at levels $h_i$ and $h_j$. The soundness is guaranteed iff:

$$\forall x \in \text{BC}_l^{h_i}, \exists y \in \text{BC}_k^{h_j} : x \subseteq \gamma(y) \tag{1}$$

There are many consensus mechanisms which are currently in use by blockchains, including PoW (Proof of Work), PoS (Proof of Stake), DPoS (Delegated Proof of Stake), PBFT (Practical Byzantine Fault Tolerance), PoB (Proof of Bandwidth), PoET (Proof of Elapsed Time), PoA(Proof of Authority) and many more [44–46] to validate blocks. Two most popular blockchain systems—Bitcoin and Ethereum—use the PoW mechanism. Due to huge energy consumption in PoW, Ethereum is now trying to adopt the alternative PoS mining approach. Moreover, Ethereum also incorporates PoA mechanism (i.e., Kovan public test chain). The existing consensus algorithms are powerful enough to deal with the situation in presence of nodes-/links-failure or even in presence of Byzantine faults. In case of our proposed multilevel blockchains systems, existing consensus algorithms can easily be adopted. In fact, the algorithm will prefer to adopt a mechanism to change leaders participating across the levels in the hierarchy, in case of suspicious behaviours of the current leaders.

### 5.4. Blockchain in Use

Given a blockchain at level $h$ which contains a number of numerical values in its blocks, one can easily lift these information to an abstract form representing their properties of interest at level $h + 1$. For example, a set of numerical values can be replaced by an interval as defined by the Galosis connection $\langle \mathsf{L}_c, \alpha_{\mathbb{I}}, \gamma_{\mathbb{I}}, \mathsf{L}_{\mathbb{I}} \rangle$. Given an abstract blockchain $BC_A$ at level h+1 , the verification operation for an element $x$ in $BC_A$ is defined below:

$$\mathsf{Search}(x, BC_A) = \begin{cases} \textit{True} & \text{if } \exists \, [x, x] \in BC_A \\ \textit{False} & \text{if } \forall \, [l_i, h_i] \in BC_A : x < l_i \vee x > h_i \\ \textit{Maybe} & \text{Otherwise} \end{cases}$$

Observe that, as the abstraction is sound, the possible results of Search operation always guarantees the soundness. That is, if the Search operation returns "*False*" in the higher level, there exist no transaction in the lower level satisfying the the same property. Formally,

$$\forall BC \in \gamma(BC_A) : \mathsf{Search}(x, BC) \in \gamma(\mathsf{Search}(x, BC_a)) \tag{2}$$

where $\gamma$ is the concretization function. Observe that $BC \in \gamma(BC_A)$ satisfies Equation (1). According to Equation (2), the Search operation may result into different cases, depicted in Table 1, due to imprecision introduced in the abstract domain. Observe that the last two cases never occur due to soundness of the Search operation defined in Equation (2) as per the abstract boolean lattice in Figure 10.

Observe that the verification at higher level of the hierarchy may lead to "*Maybe*" due to abstractions. However, one can traverse down the hierarchy to the lowest-level subnetwork and perform concrete-level verification.

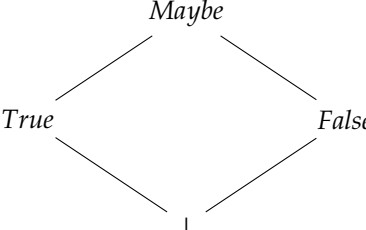

**Figure 10.** Abstract Boolean Lattice.

**Table 1.** Various Cases of Search Operation in Blockchain.

| Cases | Data Exists in Concrete Blockchain? | Data Exists in Abstract Blockchain? | Wheather Possible or Not? |
|---|---|---|---|
| 1 | *True* | *True* | *Yes* |
| 2 | *False* | *False* | *Yes* |
| 3 | *True* | *Maybe* | *Yes* |
| 4 | *False* | *Maybe* | *Yes* |
| 5 | *False* | *True* | *No* |
| 6 | *True* | *False* | *No* |

## 6. Implementation and Experimental Results

We have implemented our proposal considering a simple two-level hierachy of blockchain-networks, where three local networks $N_1$, $N_2$ and $N_3$ at level-1 are communicating with level-2 network $N_a$. A pictorial view of the system is depicted in Figure 11. We have used Ethereum client geth to build these networks where each network maintains its own blockchain separately.

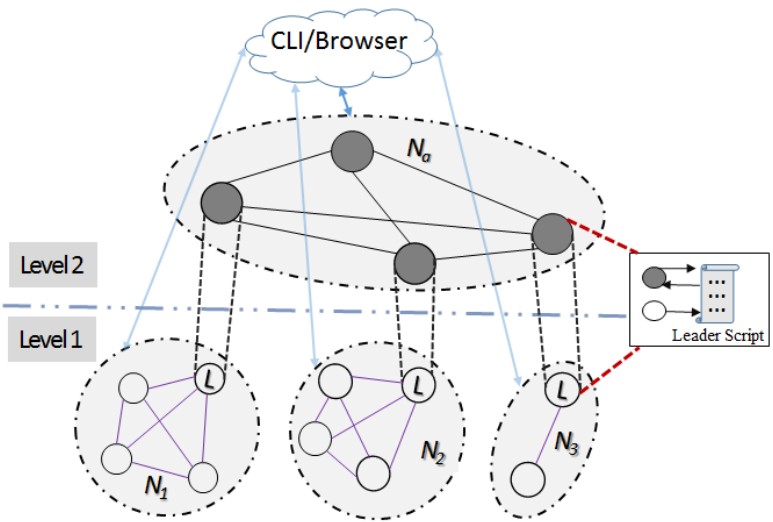

**Figure 11.** Prototypical Setup.

We have chosen a set of leaders (labelled with $L$), one from each local network $N_i$ ($i \in 1 \ldots 3$). The task of a leader is to listen a set T of its local transactions over $\delta$ wait-time and to initiate an abstract transaction $t_a$ (which contains abstraction of T) in the network $N_a$. Since our prime objective is to show the effectiveness of abstractions as a way to improve scalabality, in this experiment we have considered transactions consisting of numerical values in the form $\langle transaction_{id}, value \rangle$ pair. For abstraction, we have considered the Galois connection between numerical values domain and interval abstract domain.

The functionality of leader is implemented using JavaScript which interacts with two kinds of smart contracts, one for level-1 networks $N_i$ and other for level-2 network $N_a$. The responsibilities of these smart contracts are to initiate transactions in their respective networks and to return state information to the leader script in order to assist the abstraction process. The action performed by the leader script is depicted in Algorithm 1. Steps 5–8 are executed after every $\delta$ wait-time as per the condition specified in step 4. Observed that step 8 determines whether there exist some concrete transactions in the concrete blockchain (maintained by $N_i$) whose abstract form yet to take place in the abstract blockchain (maintained by $N_a$). Steps 6 and 7 extract information from concrete and abstract blockchains respectively with the help of their respective smart contracts. Steps 9 and 10 are responsible to apply abstraction on unattended concrete transactions and to initiate abstract transaction in $N_a$.

---

**Algorithm 1: LeaderAction**

---

    **Inputs** : Local Network ID $i$, Addresses $A_{N_i}$ and $A_{N_a}$ of Smart Contracts deployed in $N_i$ and
          $N_a$ respectively, and wait-time $\delta$

    **Output**: Abstract Transaction in $N_a$ after every $\delta$ wait-time

**1**   *startTime* = getCurrentTime();

**2**   **while** *true* **do**

**3**      *currentTime* = getCurrentTime();

**4**      **if** *currentTime* $\geqslant$ *startTime* + $\delta$ **then**

**5**         *startTime* = *currentTime*;

**6**         *concreteInfo* = getConcreteInfo($A_{N_i}$);

**7**         *abstractInfo* = getAbstractInfo($A_{N_a}$, $i$);

**8**         **if** *concreteInfo* $\notin$ $\gamma$(*abstractInfo*) **then**

**9**            *absVal* = applyAbstraction($A_{N_i}$, *concreteInfo*, *abstractInfo*);

**10**            initiateAbstractTransaction($A_{N_a}$, *absVal*);

---

For the experiment purpose, we have added more than 40,000 transactions (with values ranging from 100 to 2000) generated randomly in each local network at level-1. Table 2 depicts the experimental results which indicate the variation of blockchain-size both in concrete and abstract domains w.r.t. different wait-time ($\delta$) for leaders and different transaction-rate in local networks. The values indicate an average taken over 60 min of observation. This is important to note that since we automate the creation of local transaction randomly, there is a slight variation in the number of blocks and data-size for a given transaction-rate. The results in column 9 clearly show a significant reduction of the data-size and the number of blocks in the abstract domain w.r.t. its concrete counterpart, as we increase $\delta$. This is because, as leader waits for larger $\delta$, more number of local transaction-records are being abstracted into a single transaction. The amount of the gas consumption in the execution of both smart contracts (local and abstract agents) are shown in columns 5 and 8 respectively.

**Table 2.** Concrete vs. Abstract Blockchain.

| $\delta$ (sec) | Trans. Rate | Concrete Blockchain ($N_1 + N_2 + N_3$) | | | Abstract Blockchain | | | % of Data-Size Reduction |
|---|---|---|---|---|---|---|---|---|
| | | Data Size (bytes) | Blocks | Gas Unit | Data Size (bytes) | Blocks | Gas Unit | |
| 2 | 0.8 | 1,793,118 | 752 | 290,018,052 | 423,444 | 415 | 64,383,622 | 76.39 |
| | 1 | 2,222,252 | 802 | 376,017,792 | 455,207 | 450 | 68,488,727 | 79.52 |
| | 1.25 | 2,908,553 | 1,122 | 637,439,592 | 642,433 | 603 | 85,578,176 | 77.91 |
| 3 | 0.8 | 1,758,420 | 718 | 286,315,196 | 371,572 | 364 | 57,597,793 | 78.87 |
| | 1 | 2,238,783 | 837 | 375,500,652 | 446,487 | 442 | 65,240,401 | 80.06 |
| | 1.25 | 2,922,238 | 1,117 | 611,331,736 | 570,512 | 571 | 78,130,294 | 80.48 |
| 5 | 0.8 | 1,761,710 | 702 | 285,278,812 | 324,078 | 320 | 50,817,916 | 81.6 |
| | 1 | 2,271,361 | 941 | 381,924,264 | 433,224 | 458 | 60,367,976 | 80.93 |
| | 1.25 | 2,856,380 | 1,051 | 533,500,956 | 454,180 | 479 | 62,558,909 | 84.1 |
| 6 | 0.8 | 1,788,095 | 758 | 286,096,610 | 343,825 | 345 | 51,486,689 | 80.77 |
| | 1 | 2,284,781 | 918 | 386,398,262 | 396,471 | 406 | 54,157,749 | 82.65 |
| | 1.25 | 2,822,752 | 984 | 518,845,796 | 400,365 | 430 | 54,536,705 | 85.82 |
| 7 | 0.8 | 1,783,445 | 731 | 286,144,944 | 306,870 | 307 | 46,327,839 | 82.79 |
| | 1 | 2,245,741 | 840 | 397,938,268 | 355,289 | 384 | 48,614,951 | 84.18 |
| | 1.25 | 2,790,292 | 934 | 509,479,548 | 356,596 | 378 | 49,091,622 | 87.22 |

A sample instance occurred during our experiment is shown in Table 3 which describes how the leader performs the abstraction of local blockchain transactions. The first column of every row represents the list of transactions mined into local blockchain for $\delta = 5$ s. The second column represents its equivalent abstract form in the form of interval which is generated by the leader in network $N_1$. Observe that given a list of transactions {"32134": 138, "32135": 636, "32136": 1365} and its corresponding abstract form ["32134", "32136"]: [138, 1365], both the operations Search(636, [138, 1365]) and Search(640, [138, 1365]) result into "*Maybe*" according to its definition in Section 5.4.

**Table 3.** Concrete transactions and their abstract forms.

| Sample Transactions Instances Mined in $N_1$ for $\delta = 5$ s and Trans. Rate = 1.25 | Abstract Transaction |
|---|---|
| {"32129": 773, "32130": 485, "32131": 1218, "32132": 1452, "32133": 963} | ["32129", "32133"] : [485, 1452] |
| {"32134": 138, "32135": 636, "32136": 1365} | ["32134", "32136"] : [138, 1365] |
| {"32137": 1231} | ["32137", "32137"] : [1231, 1231] |
| {"32138": 1491, "32139": 1910, "32140": 1653, "32141": 1007, "32142": 1474, "32143": 701, "32144": 467} | ["32138", "32144"] : [467, 1910] |
| {"32145": 1770, "32146": 826, "32147": 1645, "32148": 143, "32149": 276, "32150": 724} | ["32145", "32150"] : [143, 1770] |

Let us now illustrate the experimental results of the Search operations performed both in the concrete and the abstract blockchains, depicted in Table 4. In the experiment, we have randomly chosen a test dataset containing 500 transaction-records and we have categorized it into two halves: (1) data that exists in the concrete blockchain, and (2) data that does not exist in the concrete blockchain (shown in columns 3, 4, and 5). The result of Search operation on a test dataset by varying $\delta$ and transaction-rate is shown in columns 6 and 7. The values of "Exact Discovery Rate (EDR)" are computed from the transactions representing cases 1 and 2 of Table 1, whereas the values of "Partial Discovery Rate (PDR)" are computed from the transactions representing case 3 of Table 1. Observe that, as $\delta$ and transaction-rate increase, the percentage of EDR decreases due to increasing imprecision in the abstract records.

Although the current result is showing high PDR, this can easily be reduced by tuning our abstractions and by reducing wait-time $\delta$. The use of the Abstract Interpretation framework enables one to use different levels of abstractions, from coarse-grained to fine-grained, according to the need. For example, as per the abstract lattice of interval domain, an integer $x$ can be abstracted by $[x, x]$ or it can be abstracted by $[\infty, \infty]$. Moreover, the framework is powerful enough to consider various abstract domains—either relational or non-relational or their combination. In general, the lowest-level blockchains represent concrete transaction-data, while blockchains towards the higher-level by abstract values with more degree of abstractions. Therefore, as we move towards the higher level, although the performance of verification-result process increases, its effectiveness decreases accordingly. Interestingly, the framework allows one to perform verification either at appropriate hierarchy-level depending on the degree of precision one requires in the verification result or on the concrete blockchain at lowest level by traversing down the hierarchy.

**Table 4.** Verification results: Concrete vs. Abstract.

| δ | Trans. Rate | No. of Inputs for Verification | Verification Results in Concrete Blockchain(CB) | | Verification Results in Abstract Blockchain | |
|---|---|---|---|---|---|---|
| | | | No. of Inputs | Exists in CB? | % of EDR | % of PDR |
| | 0.8 | 500 | 250<br>250 | *True*<br>*False* | 25.2 | 46 |
| 2 | 1 | 500 | 250<br>250 | *True*<br>*False* | 20.2 | 48.8 |
| | 1.25 | 500 | 250<br>250 | *True*<br>*False* | 13.4 | 49.6 |
| | 0.8 | 500 | 250<br>250 | *True*<br>*False* | 19 | 48 |
| 5 | 1 | 500 | 250<br>250 | *True*<br>*False* | 14.2 | 49.4 |
| | 1.25 | 500 | 250<br>250 | *True*<br>*False* | 17.2 | 49.8 |
| | 0.8 | 500 | 250<br>250 | *True*<br>*False* | 18.2 | 48.8 |
| 7 | 1 | 500 | 250<br>250 | *True*<br>*False* | 12.8 | 49.2 |
| | 1.25 | 500 | 250<br>250 | *True*<br>*False* | 9.2 | 50 |

## 7. Discussions

Although scalability is a major concern in decentralized peer-to-peer blockchain networks, yet researchers paid little attention till the date. The existing solutions addressed this problem in different directions: pruning out the old historical data to reduce data size in the network and in the blockchain [21], improving consensus algorithm as a way to improve latency and throughput [22], categorizing nodes in the network with predefined functionalities and enabling them to work in a distributed manner [7], etc.

All the above-mentioned approaches although solve the scalability issue to some extent, however their applicability is limited to specific application domains only. For example, the proposal in [21] is designed very specifically for cryptocurrency and generation needs all historical records, whereas the proposal in [22] is mainly concentrated on the consensus algorithm, showing that improvement in scalability of the blockchain protocol is possible to the point where the consensus latency is limited solely by the network diameter and the throughput bottleneck lies only in node processing power. This is to observe that the scope of Bitcoin-NG differs from ours in the sense that we have mainly concentrated to blockchain network structure, from mesh to hierarchical, and the abstraction of the transaction-records.

A recently published work in march 2019 [28] differs from our proposal in many ways: (1) The work focus exclusively in IoT systems that prefer to keep user's key information, and hence may not fit to other scenario in general. (2) They discriminate the participants, categorizing into device-layer, fog-layer and cloud-layer nodes, with a varying computational capabilities. The task of each layer nodes are predefined. For Example, the fog-layer nodes which lie in between device-layer and cloud layer, act as security manager to record and verify transactions that include key management information in this respective domain. Although the proposal is suitable for cross domain communications respecting privacy issues, it does not address any scalability issue in presence of large-scale nodes in the peer-to-peer network. Therefore, this is indeed interesting to see if the combination of these two improves the individual results. This is our future plan.

The hierarchical model proposed in this paper can be seen a generic model where the number of levels in the hierarchy and the levels of abstractions can easily be varied based on the application need. As we already mentioned before that the approach scale the system, alongside such partial views towards higher levels of the hierarchy also facilitate to control the access of blockchains (and hence blockchain records) differently to different groups of people or processes as per their access rights. In fact, the support of policy specification deployed through smart contract can make this model more powerful.

Because of the existence of multiple local blockchains (with different views of the global data), the model enables to design Inter-organization Blockchain Communication. This can also be empowered by allowing heterogeneous local blockchain networks to form a global view along the higher level of the hierarchy establishing Cross Blockchain Compatibility [47]. There is a scope to improve reliability in our proposal by considering multiple leaders across the hierarchical-levels, treating only one leader among them as active and others as passive. On malfunctioning, a passive leader immediately changes its state and replaces the malfunctioning-leader. Observe that all nodes, except leaders, need to maintain only small-size local blockchain. The proposed approach is suitable not only for private or consortium blockchain networks, but also for public blockchain networks. However, in case of a public blockchain network, the following challenges need to be addressed: first, since there exist multiple sub-networks across the hierarchy, there is a need to device a joining-policy for a node. This can be resolved by categorizing nodes based on their IP address (to identify local network), expected role in the network (to identify the hierarchy-level), etc. Second, as all nodes are treated equally in public network, another critical issue is how to elect leaders who will form the bridge between networks in two consecutive levels in the hierarchy. This may be resolved by adopting a suitable consensus algorithm as per the requirements. This is worthwhile to note that traditional public blockchain network is a specialized form of our model with single-level hierarchy.

## 8. Conclusions

In this paper, we introduced a hierarchical blockchain model powered by the Abstract Interpretation framework. The primary objective of the proposal is to scale blockchain network, making it suitable to fit into real-world organizational structures and providing a controlled environment for blockchain data. The results obtained by our experiment show how the variation of wait-time ($\delta$) and transaction-rate affect the size reduction of blockchain when moving from concrete to abstract domains. The verification result is also influenced by the degree of abstraction that we apply across the hierarchy. We are now working on its possible extension to empower the model with access control policy specification deployable through smart contracts, covering both static and dynamic access-control settings. On the other hand, we'll focus on the consensus mechanism to further improve the efficiency of the proposed model.

**Author Contributions:** Conceptualization, A.C.; Formal analysis, R.H.; Software, S.S. and A.M.F.; Validation, S.S. and A.M.F.; Writing—original draft, S.S. and A.M.F.; Writing—review & editing, A.C.

**Funding:** This work is partially supported by CINI Cybersecurity National Laboratory within the project FilieraSicura. The second author is supported by Visvesvaraya PhD Scheme, Ministry of Electronics and Information Technology (MeitY), Government of India.

**Conflicts of Interest:** The authors declare no conflicts of interest.

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
