# Peer review of "A Hierarchical and Abstraction-Based Blockchain Model"

_applsci, doi:10.3390/app9112343_

Round 1
Reviewer 1 Report
Summary:
In this paper, the authors proposed a model of hierarchical blockchain based on a mathematical method, the Abstract Interpretation framework. The model is proposed for scaling blockchain network and imposing access control. The concrete data contained in the low level is abstracted by the mathematical framework and saved in higher level. Lower level blockchain networks are connected with each other by leader nodes which participate in two networks. An experiment is conducted for this model and demonstrate the model is efficient for scaling. The advantage of this model is that the number of hierarchy level can be modified in order to applied in different scenario.
Strength:
+ The paper is well written and organized, and the contribution is described clearly by demonstrating adequate tables and figures.
+ In the experiment, various conditions are considered and the comparison of different conditions is persuasive to this research.
+ The authors give detailed formulas when explaining the hierarchy, make the proposed model clear to understand.
Suggestions to improve this paper:
- There are still some spelling mistakes, so authors should check throughout the paper carefully.
Author Response
See attached response

Reviewer 2 Report
This paper explores hierarchical block chain model using Abstract Interpretation framework. There are many articles on hierarchical block chain model. It is interesting that this research uses
Abstract Interpretation. However, the paper has several issues. It lacks clarity and justification of the approach. There are many mathematical aspects and most of the paper is devoted on comparing Abstract versus Concrete models. It is not clear how these are applied to insurance system as indicated in the paper. How the experimental data for insurance system was obtained? The figures do not show the layers of hierarchy. The table showing Verification Results in Abstract Blockchain have very high FDR and PDR which does not suit real-life implementation of block chain systems.
In addition, sentences are very complex. For example, the second sentence in abstract.
Overall, it is recommended to make major revisions to improbe the clarity and quality of the paper.
Author Response
See the attached Response

Reviewer 3 Report
(1) Basically, blockchain is fully connected p2p networking system. Public is very slow, but private or consorcium is faster than public. You menyioned your method is limited to private or consorcium. If you reduce data size using abstract blockchain, what about the processing speed? Please detail.
(2) Why your mehod is not appropriate in public blockchain?
(3) How much upgrade the scalability in blockchain using your method? What is the effectiveness?
(4) In table 2, what kind of examples are the case 3 and 4? Why case 5 and 6 are impossible?
Author Response
See the attached Response

Round 2
Reviewer 2 Report
Satisfied with the revisions.
Author Response
The reviewer said "Satisfied with the revisions."
We thank her/him again for the valuable suggestions.
Reviewer 3 Report
(1) Table 4. I recommend to change the marks not to confuse. For example, O pr X ..... At first glance, it is similar.
(2) In references, some papers' publication years (2016...) are bold, some are not. Please unify.
(3) Finally, please check your paper's logic again.
Author Response
see attached notes
